# Intrinsic Dimension Dynamics in Active Learning: A Geometric Diagnostic of Acquisition Behavior

**Poojith Thummala & Mohamed Abdelrazek**
Applied Artificial intelligence initiative
Deakin University
Victoria, Australia
{s219110279,mohamed.abdelrazek}@deakin.edu.au

## Abstract

Active learning reduces annotation cost by selectively querying data, yet existing evaluations rely almost exclusively on predictive performance, offering limited insight into how different acquisition strategies shape the labeled set. We study the behavior of active learning strategies through the lens of data geometry, using intrinsic dimensionality (ID) as a representation-level diagnostic. We analyze local and global intrinsic dimension statistics of samples selected by uncertainty-based (uncertainty, margin, DBAL), coverage-based (CoreSet, ProbCover), and random strategies across learning episodes, and we further conduct controlled ID-based acquisition schedules as a diagnostic stress test. To isolate acquisition effects from representation learning, we conduct the analysis in a fixed self-supervised. Across the settings we evaluate, uncertainty-based strategies tend to select samples from higher estimated intrinsic-dimensional regions, while coverage-based strategies tend to yield labeled sets with lower and more stable estimated global intrinsic dimension. Consistent with these trends, the ID-based schedules show that prioritizing low-ID samples early is substantially more effective than acquiring high-ID samples early. Overall, ID serves as a simple geometric diagnostic that complements accuracy-based evaluation.

## 1 Introduction

Active learning (AL) reduces annotation cost by selectively querying informative samples from an unlabeled pool Settles (2010); Dasgupta (2011). A wide range of acquisition strategies has been proposed, including uncertainty-based methods that target ambiguous predictions and coverage-based methods that emphasize representativeness Settles (2010); Huang et al. (2014); Yehuda et al. (2022); Sener & Savarese (2018). While these approaches exhibit distinct learning dynamics, their behavior is typically evaluated almost exclusively through predictive performance. Accuracy curves indicate whether a strategy works, but provide limited insight into how labeled sets evolve or why different strategies succeed or fail under varying conditionsTifrea et al. (2023). From a geometric perspective, active learning can be viewed as a process of progressive exploration of a data manifold Dasgupta (2011); Sener & Savarese (2018). Each acquisition decision determines which regions of the manifold are sampled and how the labeled set grows in structural complexity. Despite this interpretation, most active learning analyses do not explicitly examine how acquisition strategies interact with the geometry of the data, nor how the structure of the labeled set changes over learning episodes.

In practice, most acquisition strategies are geometry-agnostic: uncertainty-based methods operate on model output distributions rather than representation geometry, and thus may concentrate sampling in localized regions associated with ambiguity (e.g., near model decision boundaries). This can leave large regions of the data support underexplored, potentially limiting generalization to underrepresented or out-of-distribution variations Karamcheti et al. (2021). In our experiments, we observe corresponding geometric differences in the acquired sets when measured via ID. An important open question concerns the consequences of such geometry-agnostic sampling for model generalization beyond standard test accuracy. In particular, models trained on labeled sets that incompletely

cover the data manifold may exhibit fragility when confronted with valid but underrepresented variations Karamcheti et al. (2021); Amsaleg et al. (2017); Tifrea et al. (2023). Addressing this question rigorously requires generative approaches capable of probing model behavior across the full manifold, rather than relying on fixed empirical test sets Carlsson (2009); Wasserman (2018); Birdal et al. (2021). In this work, we propose *intrinsic dimensionality* as a geometry-grounded diagnostic for characterizing active learning behavior. ID characterizes the effective degrees of freedom of data in a representation space and provides a measure of both local and global geometric complexity. Rather than using ID to design new acquisition strategies, we use it as a descriptive signal to analyze how different strategies shape the geometry of the labeled set over time. To isolate geometric effects from representation learning dynamics, we conduct our analysis in a fixed self-supervised embedding space obtained from pretrained SimCLR models Chen et al. (2020), and contrast it with raw input space. By keeping representations frozen throughout AL, we interpret observed changes in ID as arising from acquisition decisions rather than evolving representations. We analyze representative uncertainty-based strategies Rigau et al. (2003); Balcan et al. (2007), coverage-based strategies (CoreSet, ProbCover) Yehuda et al. (2022), and random sampling across datasets and architectures.

**Contributions.**

1. **A simple geometric diagnostic for AL.** We formulate ID as a label-free, representation-level diagnostic for analyzing how AL strategies traverse data manifolds over rounds.

2. **Sample geometry matters: evidence via intrinsic dimension.** Across the datasets and architectures we evaluate, both standard AL runs and controlled ID-based acquisition schedules show that *which kinds of samples are labeled* (e.g., low- vs. high-ID regions) can strongly shape learning trajectories.

To the best of our knowledge, ID has not been widely used as a diagnostic to summarize how AL acquisition choices shape the geometry of the labeled set over rounds.

## 2 Intrinsic Dimensionality as a Geometric Diagnostic

Several geometric descriptors beyond ID could, in principle, be used to analyze active learning behavior, including curvature Amsaleg et al. (2017); Aamari & Levrard (2018), topological summaries Birdal et al. (2021), or geodesic coverage measures Sener & Savarese (2018); Yehuda et al. (2022). We deliberately do not pursue these directions because, Density-based measures often conflate sample concentration with structural complexity Houle (2017), while curvature and topological descriptors are difficult to estimate reliably in high-dimensional spaces and often introduce additional assumptions and hyperparameters Aamari & Levrard (2018); Birdal et al. (2021). Geodesic distances, though meaningful, are often the explicit optimization objectives of coverage-based acquisition strategies Sener & Savarese (2018); Yehuda et al. (2022), and using them as diagnostics would bias the evaluation.

We focus instead on *intrinsic dimensionality* (ID) as a minimal, label-free, and locally computable geometric diagnostic. ID characterizes the effective number of degrees of freedom required to describe data in a representation space, independent of the ambient dimensionality Ansuini et al. (2019); Levina & Bickel (2004a). It has been used in machine learning to analyze data complexity, representation quality, and generalization dynamics, as it captures structural properties not visible through performance metrics alone.Ansuini et al. (2019); Birdal et al. (2021); Amsaleg et al. (2017); Houle (2017).

In the context of active learning, ID enables us to evaluate how different acquisition strategies shape the labeled set, not just in terms of size, but in terms of geometric organization. Local ID captures the complexity of individual acquisitions Levina & Bickel (2004b), while global ID summarizes the overall structure induced by an acquisition trajectory Facco et al. (2017). Crucially, we do not treat ID as an estimator of true manifold dimension, but as a comparative summary statistic that enables consistent, interpretable analysis across strategies, datasets, and learning stages Houle (2017). This focused scope allows us to isolate the geometric effects of acquisition decisions while avoiding the confounds and complexity of more specialized geometric tools.

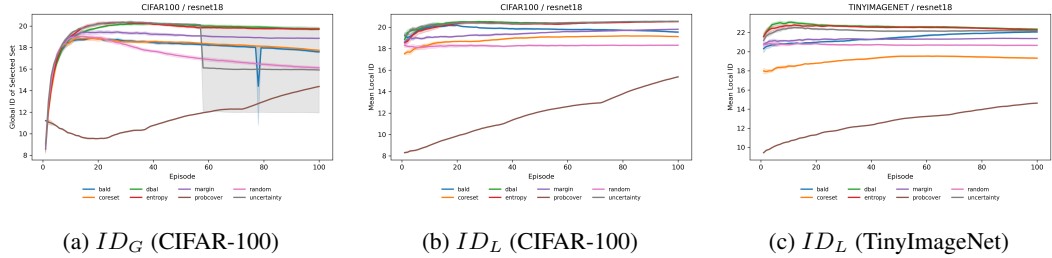

(a) $ID_G$ (CIFAR-100)  (b) $ID_L$ (CIFAR-100)  (c) $ID_L$ (TinyImageNet)

Figure 1: **ID dynamics induced by AL strategies.** *Left:* Evolution of global ID ($ID_G$) of the labeled set on CIFAR-100 using a ResNet-18 embedding. *Middle:* Mean local ID ($ID_L$) of samples acquired on CIFAR-100. *Right:* Mean local ID ($ID_L$) on TinyImageNet.

**Local Intrinsic Dimensionality**    The *local intrinsic dimensionality* (LID) of a point $x$ estimates neighborhood complexity via the Levina–Bickel MLE Levina & Bickel (2004b):

$$\mathrm{ID_L}(x) = \left[ \frac{1}{k-1} \sum_{j=1}^{k-1} \log \frac{r_k(x)}{r_j(x)} \right]^{-1}$$

where $r_j(x)$ is the distance to the $j$-th nearest neighbor. We compute these distances using the full unlabeled pool in a fixed representation space and report LID only for acquired samples. Higher LID indicates sampling from more geometrically complex or ambiguous regions.

**Global Intrinsic Dimensionality**    To summarize the structure of the labeled set $S_t$, we compute *global ID* by averaging LID scores within $S_t$, using neighbors drawn only from $S_t$:

$$\mathrm{ID_G}(S_t) = \frac{1}{|S_t|} \sum_{x \in S_t} \mathrm{ID}_L^{(S_t)}(x)$$

This captures how structurally diverse the acquired set is across acquisition rounds. We interpret $\mathrm{ID_G}$ as a diagnostic not a manifold estimator highlighting whether acquisition strategies expand into more heterogeneous regions Houle (2017).

## 3 RESULTS

**Global Geometric Evolution.**    We analyze the global ID ($ID_G$) of the labeled set on CIFAR-100. As shown in Figure 1a, acquisition strategies separate into distinct geometric regimes. Uncertainty-based strategies consistently induce higher $ID_G$, exhibiting rapid early growth followed by continued expansion. In contrast, coverage-based strategies particularly ProbCover -maintain substantially lower and more stable global ID, increasing only gradually over time. CoreSet occupies an intermediate regime, while Random sampling exhibits high variability, reflecting its lack of geometric bias.

**Local Geometric Selection.**    Analyzing the mean local ID ($ID_L$) reveals the driver of these global patterns. On CIFAR-100 (Figure 1b), uncertainty-based strategies consistently query samples with high local ID, indicating a preference for geometrically complex or ambiguous regions. Conversely, coverage-based strategies initially select low-$ID_L$ samples and only gradually expand toward higher-dimensional regions. This separation is even more pronounced in local ID than in global ID. These trends persist on the larger TinyImageNet dataset (Figure 1c), demonstrating that these geometric biases are robust to dataset scale and complexity.

**Correlation between Intrinsic Dimensionality and Accuracy.**    We observe a qualitative association between the global ID ($ID_G$) of the labeled set and the downstream test accuracy achieved by each acquisition strategy. For example, ProbCover tends to yield lower $ID_G$ across rounds and also achieves the highest test accuracies on CIFAR-100 and TinyImageNet with ResNet-18 (Table 1). Conversely, Uncertainty and DBAL tend to produce higher-dimensional labeled sets and lower final accuracies. We emphasize that this is a correlation rather than a causal claim: lower $ID_G$ may co-vary with other properties of the selected set (e.g., density, class balance, or coverage of frequent

Table 1: Test accuracy (%) at rounds 30, 60, and 100 using ResNet-18, 50 samples per round. Best results per round are shown in **bold**.

| Method | CIFAR-100 | | | TinyImageNet | | |
|---|---|---|---|---|---|---|
| | R30 | R60 | R100 | R30 | R60 | R100 |
| CoreSet | 13.94 | 18.25 | 23.26 | 5.66 | 7.49 | 8.77 |
| DBAL | 12.14 | 17.50 | 21.92 | 4.48 | 5.44 | 6.87 |
| Margin | 12.72 | 18.20 | 22.83 | 4.88 | 6.65 | 7.99 |
| ProbCover | **17.58** | **23.84** | **28.15** | **10.18** | **13.67** | **13.57** |
| Random | 13.99 | 18.80 | 23.27 | 5.66 | 7.40 | 8.99 |
| Uncertainty | 11.19 | 15.86 | 19.51 | 3.94 | 5.00 | 6.70 |

modes). Finally, to assess whether ID provides actionable information beyond post-hoc analysis, we conducted controlled acquisition experiments in which samples are ordered by their estimated ID. We emphasize that ID is not used as a standalone acquisition objective. Instead, these experiments illustrate the consequences of allocating labeling budget to different geometric regimes. Consistent with our diagnostic findings, prioritizing high-ID samples early leads to inefficient learning, while deferring high-ID regions until later stages yields substantially better performance.

## 4   DISCUSSION

**The geometric dichotomy of active learning.**    Our results suggest that active learning strategies do not merely select samples with different predictive values; they induce measurably different geometric profiles in the labeled set. Results from both standard AL curves and controlled ID-based acquisition schedules show that *which kinds of samples are acquired* can matter as much as *how many* samples are acquired: different selection biases lead to systematically different learning trajectories. Although predictive accuracy indicates whether a strategy ultimately succeeds, it collapses the acquisition process into a single outcome and obscures how learning unfolds over time. In contrast, intrinsic dimensionality makes the geometric structure of acquisition explicit. We observe a recurring separation between uncertainty-based and coverage-based methods: uncertainty-driven strategies tend to prioritize samples with higher local ID ($ID_L$), concentrating labeling budget on geometrically complex regions early, while coverage-based strategies anchor the labeled set in denser, lower-dimensional regions before expanding outward. Under the ID diagnostic, this results in more stable estimated global ID in earlier rounds and explains inefficiencies and early saturation effects that are invisible to accuracy curves alone.

**Implications for Robustness**    The tendency of uncertainty-based strategies to focus on higher estimated intrinsic-dimensional regions is consistent with the concern raised in our Introduction: geometry-agnostic sampling may leave large portions of the data support underexplored. While these strategies can refine decision boundaries, the resulting labeled sets may underrepresent lower-complexity regions that are nevertheless valid. We treat this as a hypothesis suggested by the ID diagnostic, rather than a demonstrated robustness claim. By contrast, the more stable estimated IDof coverage-based methods is consistent with more uniform exploration, which may contribute to robustness.

**Implications for acquisition design and budget allocation**    IDalso provides a lightweight, model-agnostic signal that can be computed and interpreted during the active learning loop. This creates two practical opportunities. First, ID can serve as a *real-time diagnostic* to monitor where the labeling budget is being spent (e.g., whether the method is prematurely allocating budget to high-complexity regions), enabling simple rules for budget allocation such as deferring high-ID acquisitions until later rounds. Second, ID can inform the design of new acquisition strategies by explicitly incorporating geometric preferences, for example, as a regularizer or constraint that trades off uncertainty with a desired exploration of low- vs. high-ID regions. Finally, tracking ID offers an additional axis for evaluating robustness: two strategies with similar accuracy may induce very different ID trajectories and thus different geometric coverage properties.

**Limitations and Future Work**    To isolate the geometric effects of acquisition from representation learning dynamics, our analysis leveraged fixed self-supervised embeddings. While this provides a controlled diagnostic environment, it does not capture how these geometric regimes might interact

with or distort the representation space during end-to-end training. Furthermore, while we have demonstrated the geometric consequences of acquisition (e.g., incomplete manifold coverage by uncertainty methods), rigorously quantifying the downstream impact on model fragility remains an open challenge. As standard test sets are fixed, they may not fully reveal the deficits caused by incomplete manifold coverage. Future work will focus on designing generative evaluations capable of probing model behavior across the full data manifold to better assess the link between geometric coverage and generalization robustness.

## 5 CONCLUSION

We studied active learning through a geometric lens, using IDas a simple diagnostic of the samples selected by different acquisition strategies. Across datasets and backbones (in a fixed SimCLR embedding space), we observed consistent differences in the geometry of acquired sets: uncertainty-based methods tend to query higher-ID regions, while coverage-based methods yield lower and more stable ID trajectories. Controlled ID-based acquisition schedules further show that the order in which geometric regimes are labeled can substantially affect learning dynamics, reinforcing that active learning performance depends not only on labeling budget, but also on *which* samples that budget is spent on.

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

## A APPENDIX

## A ID-BASED SAMPLING AS A DIAGNOSTIC SIGNAL

### A.1 MOTIVATION.

The purpose of this experiment is not to propose intrinsic dimensionality (ID) as a new acquisition function, but to validate its role as a diagnostic signal for interpreting and regulating labeling budget in active learning. If ID meaningfully characterizes the type of samples selected by an acquisition strategy, then deliberately allocating budget to samples from specific ID regimes should expose clear and systematic consequences. Such controlled experiments allow us to assess whether labeling samples from geometrically complex regions too early can lead to inefficient learning, despite the samples being valid.

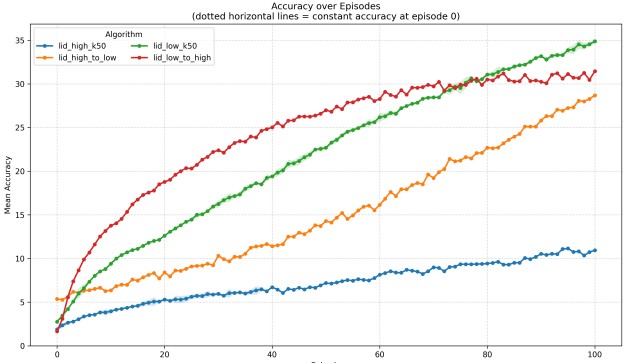

Figure 2: **Accuracy under ID-based acquisition schedules** ($k = 50$). A low-to-high ID sweep achieves rapid early learning but saturates as acquisition progresses into high-ID regimes. The inverse high-to-low sweep exhibits slow initial learning followed by a sharp performance increase once low-ID regions are sampled. Selecting exclusively high-ID samples leads to poor performance throughout training. These results illustrate that ID captures when samples become useful during active learning, rather than serving as a direct measure of immediate informativeness.

## A.2 EXPERIMENTAL SETUP.

We construct controlled acquisition schedules by ranking unlabeled samples according to their estimated local IDand selecting samples following predefined orders. Specifically, we consider three settings: (i) a round-wise sweep from low to high ID, (ii) a round-wise sweep from high to low ID, and (iii) selection restricted to high-ID samples only. IDis estimated using the kNN-based maximum likelihood estimator with a fixed neighborhood size of $k = 50$. All experiments follow the same active learning protocol as in the main paper and are conducted in a fixed self-supervised SimCLR embedding space to isolate acquisition effects from representation learning dynamics.

## A.3 RESULTS.

Figure 2 shows the resulting test accuracy trajectories. We observe a clear and consistent asymmetry in learning dynamics. The low-to-high ID schedule achieves strong early and mid-stage accuracy gains, but exhibits saturation once acquisition progresses into high-ID regimes. In contrast, the high-to-low ID schedule displays slow initial learning, followed by a sharp increase in performance once low-ID regions are eventually sampled. Selecting exclusively high-ID samples leads to consistently poor performance and early stagnation.

## A.4 INTERPRETATION.

These observations indicate that ID captures the temporal role of different geometric regions in the learning process. Low-ID regions tend to correspond to structurally coherent and representative areas of the data support that are particularly valuable for early-stage learning and model stabilization. High-ID regions, while not inherently uninformative, appear to contribute primarily to later-stage refinement and require a sufficiently trained model to be useful. As a result, allocating labeling budget to high-ID samples too early leads to inefficient learning, despite the validity of the selected samples.

## A.5 IMPLICATIONS.

These experiments substantiate the use of ID as a diagnostic signal rather than as a direct acquisition objective. By estimating ID prior to annotation, one can anticipate whether an acquisition strategy is selecting samples that are likely to support early learning or prematurely targeting complex geometric regions. This perspective helps explain the observed behavior of uncertainty-based strategies and highlights how ID can be used to identify potential waste of labeling budget without modifying the underlying acquisition mechanism.

### A.6   Key takeaways.

First, ID should not be interpreted as a standalone measure of sample informativeness. Second, ID provides a lightweight, label-free signal that characterizes when different samples are likely to be useful during active learning. Third, observing ID distributions enables informed decisions about whether to label, defer, or re-rank samples, improving budget efficiency without introducing new acquisition heuristics.

## B   Robustness to neighborhood size in intrinsic dimension estimation

### B.1   Motivation

Our intrinsic dimensionality (ID) diagnostics are computed using a $k$-nearest-neighbor (kNN) estimator. Because the neighborhood size $k$ controls the geometric scale at which ID is measured, it is natural to ask whether the qualitative conclusions are sensitive to this choice. Small $k$ emphasizes very local structure (higher variance), while larger $k$ averages over broader neighborhoods (lower variance, coarser scale). To verify that our observations are not artifacts of a particular scale, we evaluate multiple values of $k$.

### B.2   Experimental setup

We repeat the ID analysis under identical active learning runs while varying only the neighborhood size $k \in \{25, 75\}$ (and optionally $k = 10$). For each episode, we compute: (i) the mean local intrinsic dimension of the newly acquired samples, denoted $ID_L$, and (ii) a global intrinsic dimension summary of the labeled set, denoted $ID_G$. All other components dataset splits, backbone representations, acquisition budgets, and training protocol are kept fixed relative to the main experiments.

### B.3   Results

Figure 3 shows the ID trajectories for different $k$. As expected, increasing $k$ compresses the range of ID values due to neighborhood averaging, producing smoother curves. Critically, however, the qualitative separation between acquisition families remains invariant across $k$: uncertainty-based strategies consistently occupy higher-ID regimes, whereas coverage-driven strategies (e.g., Prob-Cover and CoreSet) remain in lower-ID regimes for substantially longer portions of the acquisition process. This ordering is preserved for both $ID_L$ and $ID_G$.

### B.4   Interpretation

The persistence of these trends under large neighborhoods (e.g., $k = 75$) indicates that the observed differences cannot be explained solely by highly local effects (such as small-scale density fluctuations). Instead, the ID diagnostics capture stable, strategy-dependent exploration of geometric regimes in representation space: coverage-based methods preferentially acquire samples from lower-complexity regions before expanding outward, while uncertainty-based methods transition more quickly into higher-complexity regions. Therefore, the ID-based conclusions in the main paper are robust to substantial changes in the estimator scale.

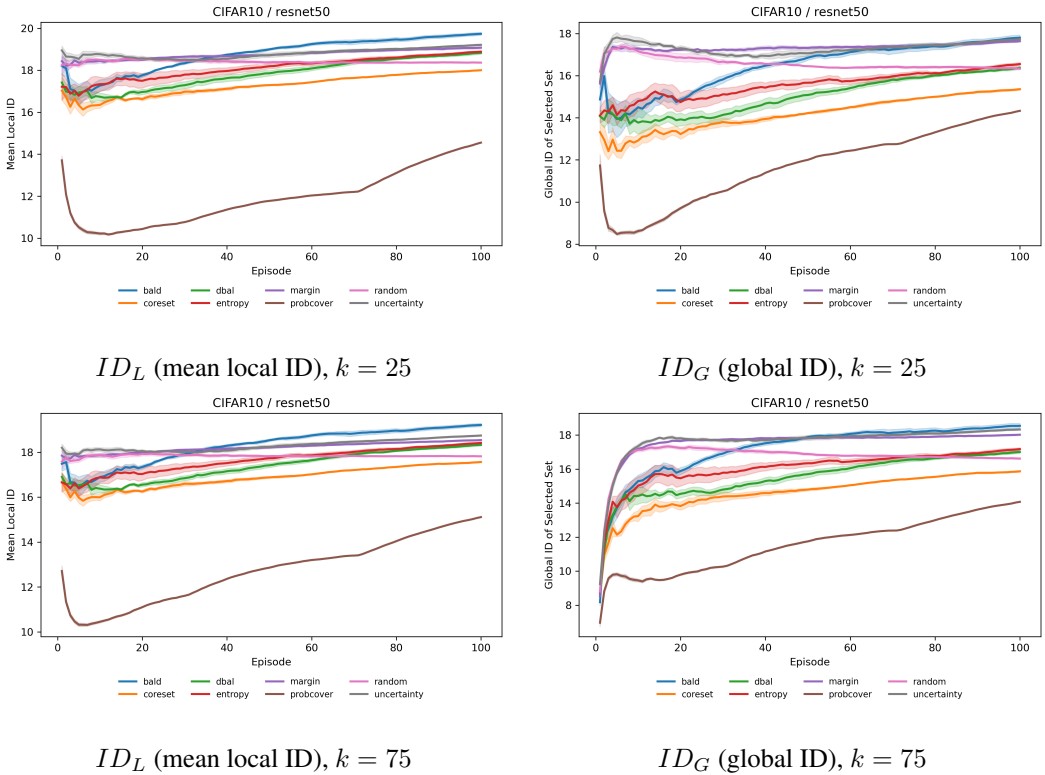

$ID_L$ (mean local ID), $k = 25$                     $ID_G$ (global ID), $k = 25$

$ID_L$ (mean local ID), $k = 75$                     $ID_G$ (global ID), $k = 75$

Figure 3: **Robustness of ID diagnostics to neighborhood size** $k$**.** Increasing $k$ smooths and compresses ID values as expected, but the qualitative ordering and separation between uncertainty-based and coverage-based acquisition strategies remains invariant.

