# OpenReview forum: "INTRINSIC DIMENSION DYNAMICS IN ACTIVE LEARNING: A GEOMETRIC DIAGNOSTIC OF ACQUISITION BEHAVIOR"
_ICLR.cc/2026/Workshop/GRaM — ICLR 2026 Workshop GRaM Poster_

### Official Review · Reviewer_uPRW · 2026-02-08

**Rating:** 7
**Confidence:** 3

**Review:**

__Overview__
This paper analyzes active learning strategies through the lens of data geometry, specifically using Intrinsic Dimensionality (ID) as a diagnostic tool. The authors observe that uncertainty-based strategies systematically select samples from high-ID regions (complex or ambiguous areas), while coverage-based strategies select low-ID samples (representative areas). Controlled experiments where samples are acquired based on their ID ranking suggest that acquiring low-ID samples early is significantly more effective for model performance than prioritizing high-ID samples.

__Strength__
- Novel Diagnostic: The work moves beyond standard accuracy-vs-budget plots to explain why strategies behave differently based on the geometric properties of the selected data.
- Clear Empirical Evidence: The results establish a consistent geometric distinction between uncertainty-based and coverage-based methods across different datasets.
- Validation: The controlled experiments using ID-based schedules provide a compelling argument that the curriculum of sample complexity (low to high ID) directly impacts learning efficiency.

__Weakness / Question__
- Fixed Representations: The analysis relies on fixed self-supervised embeddings to isolate acquisition effects. In practical active learning, the model representation evolves with the labeled set. It remains unclear if these geometric trends hold when the embedding space changes during training.
- ID vs. Density: Low intrinsic dimension often correlates with high data density. The paper could better clarify whether ID provides unique insights that distinguish it from simple density-based diagnostics.

__Relevance to GRaM topics__
Yes

__Originality and Novelty__
The application of ID dynamics as a diagnostic tool for active learning is original. It offers a new perspective on the behavior of standard acquisition strategies rather than simply proposing a metric optimization.

__Technical Soundness__
The methodology is sound. The use of the Levina-Bickel estimator is standard, and the experimental setup effectively isolates the variables of interest.


__Clarity and organization__
The writing is concise and easy to follow. The distinction between local and global ID is well-explained.

__Prior work and PMLR suitability__
N/A

__Double-blind compliance__
The paper adheres to double-blind review standards.

__LLM usage and guideline compliance__
The writing style is natural and precise. There are no signs of excessive or improper LLM generation.

**Pmlr Suitability:**

NA

---

### Official Review · Reviewer_rw55 · 2026-02-18

**Rating:** 7
**Confidence:** 3

**Review:**

Overview: This research explores active learning via geometric insight, using intrinsic dimensionality as a diagnostic to analyze how different acquisition strategies shape the geometry of the labeled set over time. Through isolating acquisition effects in fixed representation spaces and validating the findings with controlled ID-based acquisition schedules, the work provides clear geometric findings into learning dynamics beyond standard accuracy-based evaluation.

Strengths
-The short paper provides a clear and geometry-grounded diagnostic framework for analyzing active learning behavior using intrinsic dimensionality
-Offers a novel perspective on active learning by characterizing how acquisition strategies shape the geometry of the labeled set over time, rather than evolving strategies solely through accuracy.
-Experimental results are consistent across datasets and architectures, supporting the robustness of the conclusions.

Weaknesses
-The analysis is diagnostic rather than prescriptive, the paper doesn't propose new acquisition strategies that directly leverage intrinsic demography
-The relationship between intrinsic dimensionality and downstream robustness or generalization beyond standard test sets remains largely correlational
-While the geometric analysis is compelling, some practical implications for real-world active learning systems could be further elaborated.

**Pmlr Suitability:**

NA

---

### Official Review · Reviewer_hecs · 2026-02-24
**Review of Submission52**

**Rating:** 6
**Confidence:** 2

**Review:**

The authors propose ID as a "label-free, representation-level diagnostic" to characterize the behavior of uncertainty-based versus coverage-based AL strategies. By conducting experiments in a fixed self-supervised embedding space, they isolate the effects of acquisition decisions from the complexities of representation learning.

### Strengths

The paper’s premise is highly relevant and fits the scope of a workshop like GRaM. Specifically:

- **Novel Perspective:** Moving beyond predictive performance to look at the **geometric evolution** of the labeled set offers a fresh way to interpret why certain AL strategies saturate or fail.
- **Actionable Insights:** The finding that low-ID samples support model stabilization early while high-ID samples are better suited for later-stage refinement provides a practical heuristic for budget allocation.
- **Robustness Checks:** The authors include an appendix demonstrating that their qualitative findings are relatively robust to the choice of neighborhood size () in the ID estimator.


### Weaknesses & Lack of Rigor

The primary flaw of the paper is the reliance on **qualitative observation** over **statistical validation**.

- **Absence of Correlation Statistics:** The authors claim a "qualitative association" between ID and accuracy. However, they provide no formal metrics—such as **Spearman’s rank correlation** or **Pearson’s correlation**—to quantify this relationship across the various methods and rounds. Without these, the claim that lower ID "co-varies" with higher accuracy remains speculative.
- **Methodological Unreliability:** Comparing the visual ordering of curves in Figure 1 against the numerical values in Table 1  is not a scientifically rigorous way to present results. The "separation into distinct geometric regimes" should be supported by statistical tests (e.g., significance tests for differences in mean ID across acquisition families).
- **Over-reliance on Visual Interpretation:** The conclusion that coverage-based methods are more "stable"  is based on the visual "smoothness" of the curves. Quantifying this stability through variance measures or rate-of-change statistics would significantly strengthen the argument.

### **Recommendation**

The paper is **interesting and worth a space in the workshop**, as it introduces a useful diagnostic framework. However, it currently feels more like an observational study than a rigorous experimental evaluation.

**Pmlr Suitability:**

NA

---

### Meta-Review · Program_Chairs · 2026-02-28

**Decision:**

Accept

**Metareview:**

The geometric approach to active learning is interesting and relevant to the workshop. Reviewers also see this work as a good fit to the gram community.

**Relevance To Proceedings:**

Tiny paper — does not apply

**Relevance To Workshop:**

Yes — suitable for GRaM

---

### Decision · Program_Chairs · 2026-03-02

Accept (Poster)